# Evapotranspiration Retrieval Using S-SEBI Model with Landsat-8 Split-Window Land Surface Temperature Products over Two European Agricultural Crops

**Vicente Garcia-Santos** *![ID], **Raquel Niclòs** ![ID] **and Enric Valor** ![ID]

Department of Earth Physics and Thermodynamics, Faculty of Physics, University of Valencia,
46100 Valencia, Spain; raquel.niclos@uv.es (R.N.); enric.valor@uv.es (E.V.)
* Correspondence: vicente.garcia-santos@uv.es

**Abstract:** Crop evapotranspiration (ET) is a key variable within the global hydrological cycle to account for the irrigation scheduling, water budgeting, and planning of the water resources associated with irrigation in croplands. Remote sensing techniques provide geophysical information at a large spatial scale and over a relatively long time series, and even make possible the retrieval of ET at high spatiotemporal resolutions. The present short study analyzed the daily ET maps generated with the S-SEBI model, adapted to Landsat-8 retrieved land surface temperatures and broadband albedos, at two different crop sites for two consecutive years (2017–2018). Maps of land surface temperatures were determined using Landsat-8 Collection 2 data, after applying the split-window (SW) algorithm proposed for the operational SW product, which will be implemented in the future Collection 3. Preliminary results showed a good agreement with ground reference data for the main surface energy balance fluxes $R_n$ and LE, and for daily ET values, with RMSEs around 50 W/m$^2$ and 0.9 mm/d, respectively, and high correlation coefficient ($R^2$ = 0.72–0.91). The acceptable uncertainties observed when comparing with local ground data were reaffirmed after the regional (spatial resolution of 9 km) comparison with reanalysis data obtained from ERA5-Land model, showing a StDev of 0.9 mm/d, RMSE = 1.1 mm/d, MAE = 0.9 mm/d, and MBE = −0.3 mm/d. This short communication tries to show some preliminary findings in the framework of the ongoing Tool4Extreme research project, in which one of the main objectives is the understanding and characterization of the hydrological cycle in the Mediterranean region, since it is key to improve the management of water resources in the context of climate change effects.

**Keywords:** evapotranspiration; S-SEBI; remote sensing; Landsat-8; split-window; LST

## 1. Introduction

The importance of evapotranspiration (ET) in the global hydrological cycle is well known, for which accurate estimates provide an effective tool for the irrigation scheduling, water budgeting, and planning of the water resources associated with irrigation in croplands [1]. Understanding the processes that govern the crop growth helps in predicting the tendency of global climate change, a key challenge to face.

ET temporal evolution can be monitored with field-based instrumentation, e.g., lysimeter, Bowen ratio energy balance system, scintillometer, eddy covariance, etc. However, these are just highly accurate local measurements, not representative of the atmospheric conditions nor the surrounding large-scale spatial heterogeneity, or at least representative of a radius of 115 m centered in the flux tower [2]. This situation makes room for the use of remote sensing techniques, which provide geophysical information at large spatial scale and over relatively long time series.

In the last 30 years, several algorithms and ET models have been proposed, improved, and validated. A thorough review of such ET models can be found in [3]. The present study focuses on one of these ET models, the so-called Simplified Surface Energy Balance Index

(S-SEBI [4]). If the hydrological conditions of the surface are extreme enough to observe a significant range of land surface temperatures (LST) for the corresponding reflectance spectra, i.e., albedo values ($\alpha_s$), then S-SEBI does not need additional meteorological data.

To our knowledge, just eight studies have dealt with the implementation and validation of the S-SEBI model to sensors on board Landsat-8. Six out of these eight studies tested the S-SEBI ET algorithm, introducing as input the LST retrieved from Landsat-8 with a single-channel method [5], at crop sites located in the USA [6,7] and Asia [8–11]. The last two references used a split-window (SW) algorithm to estimate LST, and later the S-SEBI model to obtain the ET on natural grasslands in Brazil, and a cropland in Spain [12,13].

The present study is a short communication of important preliminary findings in the framework of an ongoing research project called Tool4Extreme, which pursues the understanding and characterization of the hydrological cycle in the Mediterranean region as key to improve the water management in the context of global change. The project analyzes and quantifies the precipitation variability and temperature trends that can modify the atmospheric evaporative demand in the area. This paper shows the project's preliminary findings on ET retrievals from remote sensing techniques applied to two agricultural crops. In general terms, it is expected that water stress will increase in the future due to the higher water atmospheric demand under higher temperatures combined with similar or lower amount of annual precipitation, and it may induce significant changes in fresh water availability for agricultural, industrial, and urban uses, leading to consequences for the ecosystems.

It is also worth noting that this is the first study that applies the official SW-LST algorithm, which will soon be implemented in the Landsat-8 Collection 2, as input of the S-SEBI model to estimate the ET over two agricultural sites located in Europe. The objectives of the present study are (i) to show preliminary findings of the ongoing Tool4Extreme project in relation to the study of the hydrological cycle in a region as key to improve the water management in the context of global change, and (ii) to show the proper functioning of the split-window LST algorithm that will be implemented soon in Collection 3 of Landsat-8.

Section 2 describes the study areas and data used. The methodology applied to retrieve the satellite data is detailed in Section 3. Section 4 shows the main results and contains a discussion derived from the findings. Finally, Section 5 lists the main conclusions.

## 2. Study Areas and Data

### 2.1. Description of Sites and Field Flux Data

The two agricultural sites selected in this study (Figure 1) are integrated in the FLUXNET dataset [14]. One site is located in Oensingen in Switzerland [15] and is operated by the Swiss Federal Institute of Technology of Zurich. The second site is placed in Borgo Cioffi, Italy [16], and is maintained by the Consiglio Nazionale delle Ricerche-Istituto per i Sistemi Agricoli e Forestali del Mediterraneo.

The Oensingen site is located in the Canton Solothurn, 47.286471°N, 7.733712°E, at 452 m a.s.l. The site extension is 1.55 ha of cropland (top scenes in Figure 1). The site soil is classified as fluvisol with textural composition of 42% clay, 33% silt, and 25% sand. Mean temperature is 8.4 °C and the annual precipitation is around 100 mm evenly distributed along the year (see Figure 2). The site is cultivated with varying crop types (e.g., winter wheat, rapeseed, potato, winter barley, etc.) in a long-term crop rotation system repeated every four years. A barley crop, according to the farmer, was being cultivated at the time of this study. The climate of our Switzerland site can be considered Mediterranean, according to [17].

Borgo Cioffi farm site is located in South Italy (40.52353°N, 14.957449°E, at 15 m a.s.l.). The experimental site (bottom scenes in Figure 1) covers 10 ha and it is characterized by a typical Mediterranean climate with hot and dry summers and cool and rainy winters, with a mean annual precipitation of 45 mm, also evenly distributed along the year, except for the summer months (see precipitation distribution at Borgo Cioffi in Figure 2). In 2017–2018

the dominant crop was corn, which has a growing period of 100 days and left harvest in August–September. Field is irrigated 270 mm/year on average with a central-pivot system.

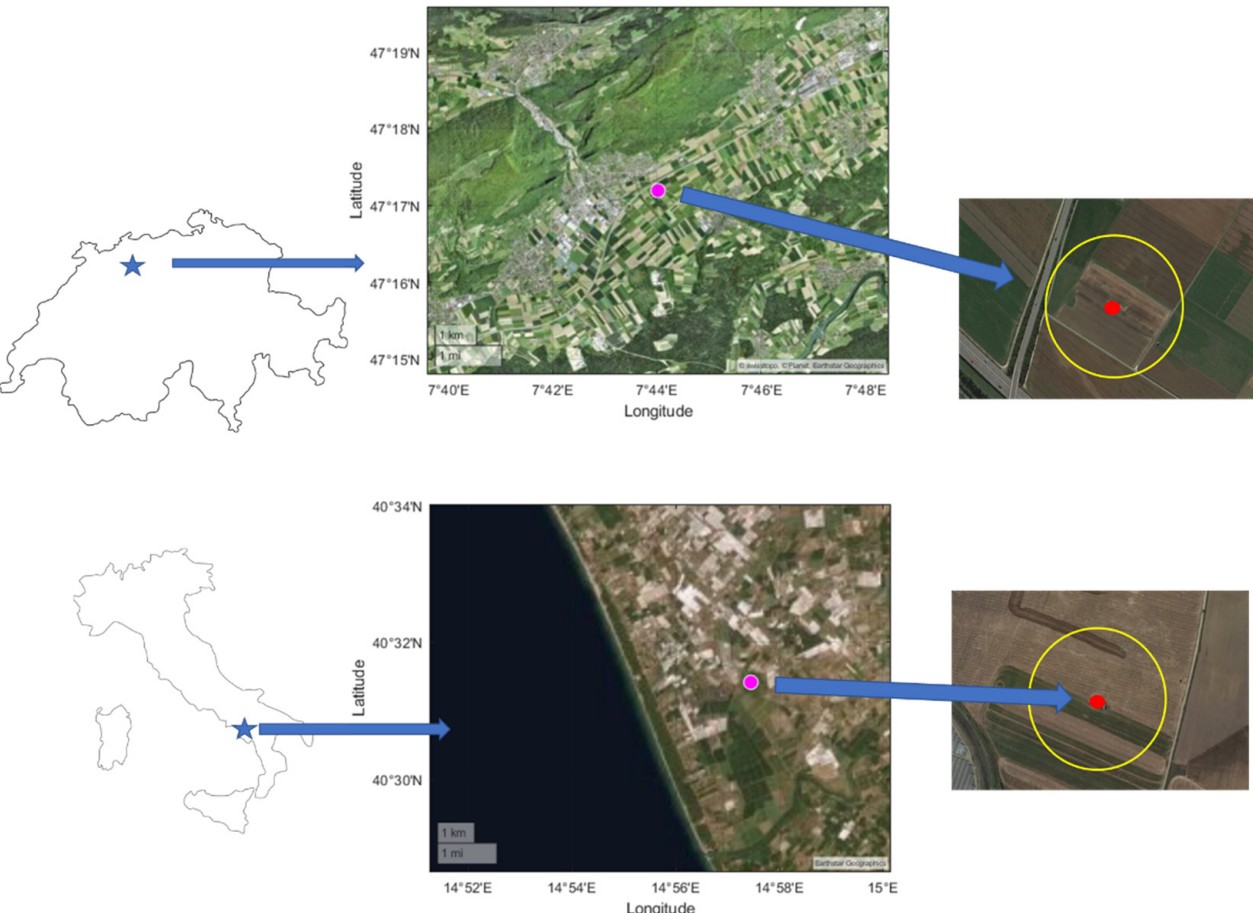

**Figure 1.** Location of the two sites in their corresponding countries (blue stars). Magenta circles in the middle scenes show the exact position of both site stations based on satellite scenes extracted from the Matlab geoaxes function. Finally, red circles in the right Google Earth images show where the stations are placed, and yellow circles show the 100 m action radius of the eddy covariance measurements according to [2].

In both cropland sites, an eddy covariance (EC) system is operating to monitor surface fluxes. $CO_2$ and $H_2O$ fluxes were measured continuously at 20 Hz resolution with a LICOR 7500 (LICOR Inc., Lincoln, NE, USA) infrared absorption spectrometer in combination with a Gill Solent R3-50 (Gill Instruments, Ltd., Lymington, UK) three-dimensional sonic anemometer. Fluxes were computed as 30 min averages, and high-frequency damping losses and apparent fluxes were also assessed. A 100 m area was considered (average of $3 \times 3$ Landsat-8 pixels, centered in the site) as a compromise between the spatial resolution of the Landsat-8 and the spatial footprint of the EC field data, since, according to [2], the EC data represent about 90% of the flux originated within a circle with a radius of 115 m centered in the flux tower.

*2.2. Remote Sensing Data*

In this study, two satellite datasets were required to estimate ET. We used Landsat-8 Operational Land Imager/Thermal Infrared Sensor (OLI/TIRS) scenes [18] processed to obtain the Normalized Difference Vegetation Index (NDVI), albedo, and LST.

A total of 32 cloud-free Landsat-8 Collection 2 scenes were downloaded from the U.S. Geological Survey service through www.earthexplorer.usgs.gov (accessed on 14 December

2021). OLI scenes were already atmospherically corrected. Just eight of these images corresponded to the Oensingen site for the 2018 year (WRS-2 path/row 195/27, 10:15 UTC acquisition time), and the other 24 scenes to Borgo Cioffi site within 2017–2018 (WRS-2 path/row 189/32, 09:40 UTC acquisition time). Landsat-8 scenes covered the whole year at both sites; two years in the case of the Italian site. However, just one (two maximum) clear scene per month was obtained at each site, which highlighted the scarcity of good-quality satellite data and the lack of continuity of suitable Landsat-8 scenes at the study areas.

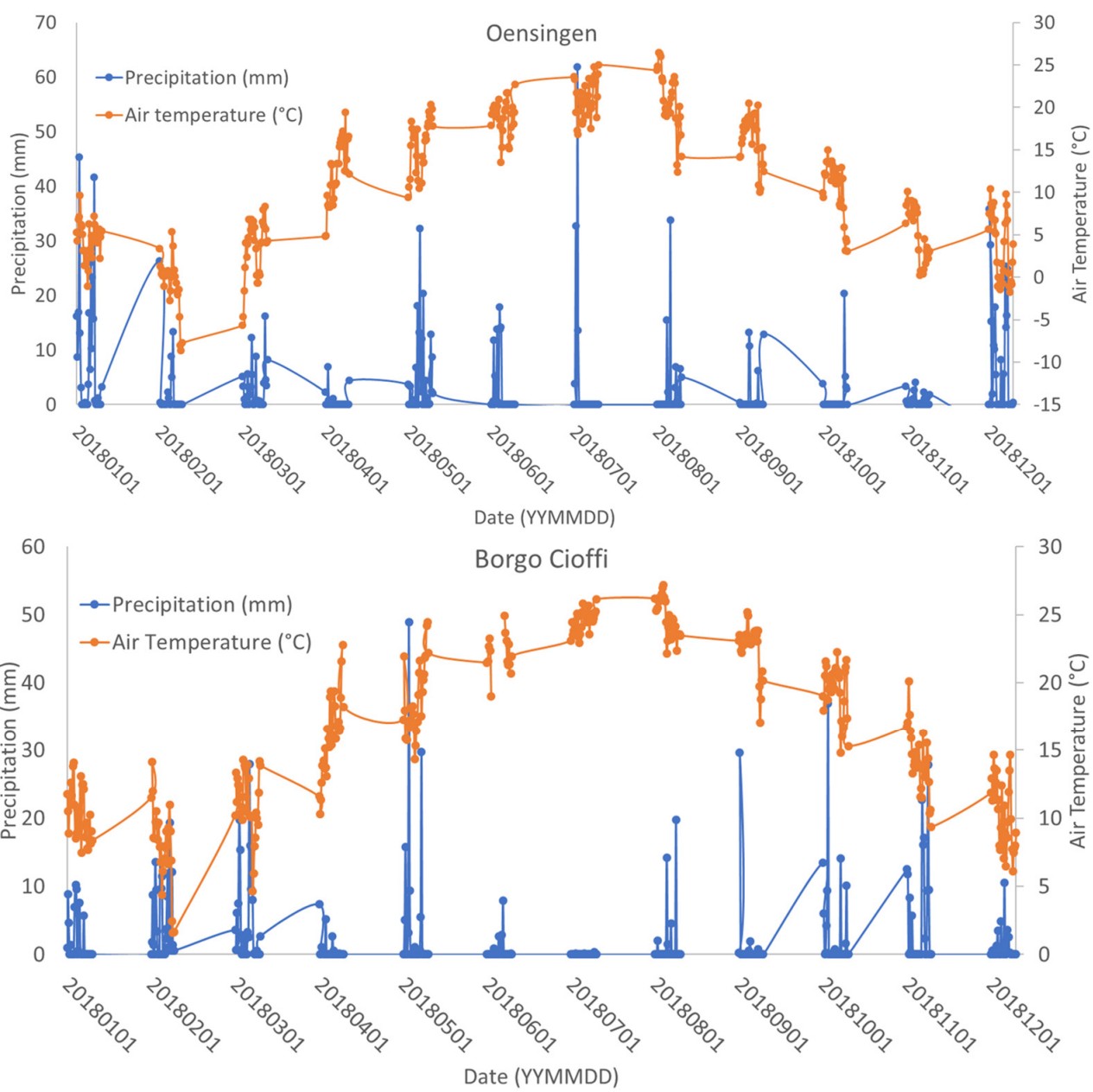

**Figure 2.** Daily mean precipitation (in mm) and air temperature (in °C) for the year 2018 at both studied sites.

## 3. Method

It is worth mentioning that all Landsat 8 products (e.g., LST, albedo, etc.) used in this study were previously preprocessed to limit the extent of the study site to a region of 20 km × 20 km centered around the coordinates of both site stations. As a consequence, derived outputs (i.e., $R_n$, LE, or $ET_d$) were also estimated for such areas.

### 3.1. S-SEBI ET Model Description

The physical-based assumption in the S-SEBI model [4] is that ET varies with LST for a homogeneous surface. For a given surface albedo and boundary layer condition, there exists a gradient of temperature differences between surface and atmosphere at some crop height (LST − $T_a$), that ranges from a maximum, where it is presumed that ET equals 0, to a minimum value, where it is presumed a potential flux according to the Penman–Monteith equation. S-SEBI has the advantage that no additional meteorological data are required if the hydrological conditions of the surface are extreme enough to observe a significant range of temperature differences for the corresponding reflectance spectra. Therefore, in the case that, for a surface albedo ($\alpha_s$) value, the two temperature difference extremes exist, a maximum value of the sensible heat flux occurs, $H_{max}$, and latent heat flux $LE_{max}$, $ET_i$, can be expressed as

$$\lambda ET_i = \frac{a_{max} + b_{max}\alpha_s - T_s}{a_{max} - a_{min} + (b_{max} - b_{min})\alpha_s}(R_n - G) \tag{1}$$

where $R_n$ and $G$ are net radiation and soil heat fluxes (W/m$^2$), $\lambda$ is the latent heat of vaporization of water, $a_{max}$ and $a_{min}$ are the slope coefficients, and $b_{max}$ and $b_{min}$ are the offsets when the maximum and minimum LSTs ($T_s$) are linearly regressed against the albedo $\alpha_s$.

### 3.2. LST and Albedo Landsat-8-Derived Products

LST is the main input to retrieve the ET with the S-SEBI model. As commented previously, most of the previous studies testing the S-SEBI model were based on LST values retrieved with a single-channel method, since shortly after the L8 launch, a radiance from outside of the TIRS's field of view produced a non-uniform ghost signal across the focal plane that varied depending on the out-of-scene content. A stray-light correction algorithm (SLCA) was developed to solve this problem [19], with accurate results according to [20,21]. The SLCA is implemented operationally into the United States Geological Survey (USGS) ground system for Landsat-8 starting in February 2017. Furthermore, the SLCA was also applied to Landsat-8 scenes prior to February 2017, and the scenes were updated.

In this study, the SW method proposed by [22] was used to estimate the LST from brightness temperatures of Landsat-8 TIRS bands 10 ($T_{10}$) and 11 ($T_{11}$), since this algorithm will be used as the next official SW-LST product for Landsat-8. The algorithm is given by the next equation:

$$T_s = b_0 + \left(b_1 + b_2\frac{1-\varepsilon}{\varepsilon} + b_3\frac{\Delta\varepsilon}{\varepsilon^2}\right)\frac{T_{10} + T_{11}}{2} + \left(b_4 + b_5\frac{1-\varepsilon}{\varepsilon} + b_6\frac{\Delta\varepsilon}{\varepsilon^2}\right)\frac{T_{10} - T_{11}}{2} + b_7(T_{10} - T_{11})^2 \tag{2}$$

where coefficients $b_k$ (k = 0–7) are sensor-dependent (and potentially water-vapor-dependent) coefficients. $\varepsilon$ and $\Delta\varepsilon$ are the surface mean emissivity and emissivity difference at bands 10 and 11, respectively. A recent validation study [21] showed that this SW algorithm estimates the LST with a root-mean-square difference (RMSD) of $\pm0.8$ K over a full vegetated surface, such as both studied here. In this study, $\Delta\varepsilon$ was fixed to a value of 0.015, according to the uncertainty associated with emissivities in the ASTER GED database [23] at 100 m, and $\varepsilon$ was established considering similar values in both bands, since the spectra are quite flat in the 10–12 μm range. $\varepsilon$ was estimated with an NDVI threshold method proposed by Zhang et al. [24].

To retrieve the broadband surface albedo (0.25–5 μm), a method proposed by [25] was implemented in this work. This method requires the at-surface spectral reflectance ($\rho_i$) measured by the Landsat-8 OLI sensor at bands 2–7, as follows:

$$\alpha_s = 0.2453\rho_2 + 0.0508\rho_3 + 0.1804\rho_4 + 0.3081\rho_5 + 0.1332\rho_6 + 0.0521\rho_7 + 0.0011 \tag{3}$$

### 3.3. $R_n$ and G Fluxes from MSG/Landsat-8-Derived Products

The difference $R_n - G$ in (1) is known as the available energy and it is the residual flux partitioned in H and LE. $R_n$ represents the total heat energy considered in the balance between incoming and outgoing short-wave (0.15 to 5 μm) and long-wave (3 to 100 μm) radiation under steady atmospheric conditions. It can be expressed as

$$R_n = (1 - \alpha_s)SWIR + \varepsilon\left(LWIR - \sigma T_s^4\right) \tag{4}$$

where *SWIR* and *LWIR* are, respectively, the short-wave and long-wave incoming radiation, reaching the surface, for which values used for this study were measured directly by the stations installed in both sites, and $\sigma$ is the Stefan–Boltzmann constant.

On the other hand, G is the heat energy used for warming or cooling the substrate soil volume and it is retrieved in this study as a fractional estimate of $R_n$, as [3]:

$$G = (0.295 - 0.01331\rho_5/\rho_4)R_n \tag{5}$$

### 3.4. Daily Evapotranspiration

Daily ET estimates were compared and validated with ground data. The remote sensing LST-based ET models retrieve instantaneous ($ET_i$) values, which are of very little use for most hydrological and water resource management applications. Therefore, it is necessary to convert $ET_i$ to at least daily estimates ($ET_d$), or even longer time intervals, to make full application of the remote sensing data. After analyzing different approximation methods proposed by several studies, we applied the equations suggested by Jackson et al. in [26]:

$$ET_d = \frac{2N}{\pi \sin\left(\frac{\pi t}{N}\right)}ET_i \tag{6}$$

where $N$ and $t$ are the number of sun hours for the whole day and since sunrise, respectively. For cloud-free days or with relatively constant cloud cover, the sine function has demonstrated to provide reliable estimates of $ET_d$.

To obtain daily ET values from ground data, we converted the half-hourly averaged LE data to half-hourly ET (mm/half-hourly). After these first steps, $ET_d$ values were obtained by adding all the $ET_{30min}$ terms per day.

### 3.5. ERA5-Land $ET_d$ Reanalysis Data

ERA5-Land is a reanalysis dataset of land variables [27] over several decades, produced by replaying the land component of the ECMWF ERA5 climate reanalysis, combining model data with worldwide ground observations, using the laws of physics and ERA5 atmospheric variables, such as air temperature, air humidity, and pressure, corrected to account for the altitude difference between the grid of the forcing and the higher resolution grid of ERA5-Land. The temporal (i.e., hourly ET values used for his study) and spatial ($0.1° \times 0.1°$, i.e., native 9 km) resolutions of ERA5-Land make this dataset very useful for all kinds of land surface applications (e.g., flood and drought forecasting).

ERA5-Land reanalysis data are used in this study for validation purposes at a regional scale, since these data have high temporal resolution in these regions, and they can be matched up with concurrent satellite data. ERA5-Land reanalysis data are also widely accepted by the ET community as a reliable source, being even considered as a replacement of regional ground reference $ET_d$ measurements [28]. The ERA5-land variable of total evaporation (TE) was used to compare these data with the daily ET values estimated from Landsat-8-based data. TE (in m of water equivalent units) is the accumulated amount of Earth's surface-evaporated water into vapor in the air above it. This variable was accumulated from the beginning of the forecast to the end of the forecast step. For the purpose of this study, ERA-Land TE variable was downloaded for the last hour of the day, and data units were converted from meters to millimeters, and the TE was finally obtained in mm/d, i.e., daily ET.

For the comparison, an upscaling of the original 30 m Landsat-8 $ET_d$ data to the native resolution of ERA5-Land TE (i.e., 9 km) was performed, and then both scenes were geo-collocated. Four pixels of the upscaled Landsat-8 scenes and ERA5-Land TE product should cover the 20 km × 20 km study region extent. However, it is obvious for the Italian site that the upper and lower left part of the scene (see bottom images in Figure 1) is covered by the sea, so just the upper and lower right parts of the scene were used for the comparison. In the case of the Swiss site, four pixels were used for the comparison, but sometimes clouds appeared in the scenes, so this number was reduced to just 2–3 pixels. The same happened for the Italian site, reducing the number of pixels to 1–2 cloud-free pixels over the study region.

*3.6. Statistical Analysis*

All instantaneous $R_n$ and LE fluxes, as well as $ET_d$ retrieved from satellite data at original 30 m resolution, were upscaled to 100 m for validation purposes, since according to Kljun et al. [2], EC station offers an effective 100 m radius estimation of fluxes measured.

All the commonly used model performance metrics were assessed for comparing validation results. Thus, the assessed statistical metrics were correlation coefficient ($R^2$), standard deviation (StDev), root mean square error (RMSE), mean absolute error (MAE), mean bias error (MBE), Nash–Sutcliffe model efficiency coefficient (NSE), and agreement index (AI). Further information about all these statistical coefficients and their expressions can be found in [29,30].

# 4. Results and Discussion

## 4.1. Comparison with Ground-Measured Data

Figure 3a shows the net radiation retrieved from satellite data using Equation (4), compared with the instantaneous one measured at the ground station of Borgo Cioffi, for 14 out of 24 dates used in this study. For the resting dates, the Italian station did not provide measured $R_n$, as was the case for the Swiss station for all dates selected. Figure 3b shows the comparison of satellite LE retrievals, using S-SEBI Equation (1), with the LE term measured at both stations for the 32 dates selected. In both cases, data are shown together with the corresponding linear regression equations and the determination coefficients ($R^2$). Good agreement between satellite data and ground station data are shown for both variables, especially for $R_n$ ($R^2 > 0.9$ in this case), with a satellite overestimation for the lowest fluxes values. Although a wider set of ground Rn data would be necessary to achieve a more statistically significant comparison results, the S-SEBI model seems to perform correctly for high fluxes values. Statistical errors can be observed in Table 1.

Figure 4 shows a comparison between $ET_d$ estimated with the S-SEBI model using Landsat-8 data and that calculated from ground LE measurements (see Section 3.4). Here, as in flux terms of Figure 3, a good correlation of satellite results with ground data is also observed, with a better fitting for high $ET_d$ and an overestimation of the S-SEBI model compared with ground $ET_d$ data. An outlier is observed for the $ET_d$ ground value of 4.8 mm/d, probably because the G flux is not catching the correct presence of vegetation in the scene. $R^2$ would increase up to 0.82 if this outlier was removed from the regression. Statistical results are also included in Table 1.

Table 1 shows the statistical results for the analysis of the ground-model differences for $R_n$ and LE fluxes, as well as $ET_d$ values, retrieved with Landsat-8 data. Results compared with field reference values presented a relatively high correlation (with $R^2$ from 0.72 to 0.91) and low uncertainties of around 40 W/m$^2$ and 0.7 mm/d (MAE), 5–8 W/m$^2$ and 0.2 m/d (StDev), and 50 W/m$^2$ and 0.9 mm/d (RMSE). This statement is reinforced by the high AI (0.996–0.997) and NSE (0.6–0.9) values. This points out, in general terms, that satellite-retrieved $R_n$ and LE values, as well as modeled daily ET, tend to underestimate ground-measured data, according to the negative MBE observed results for all the parameters.

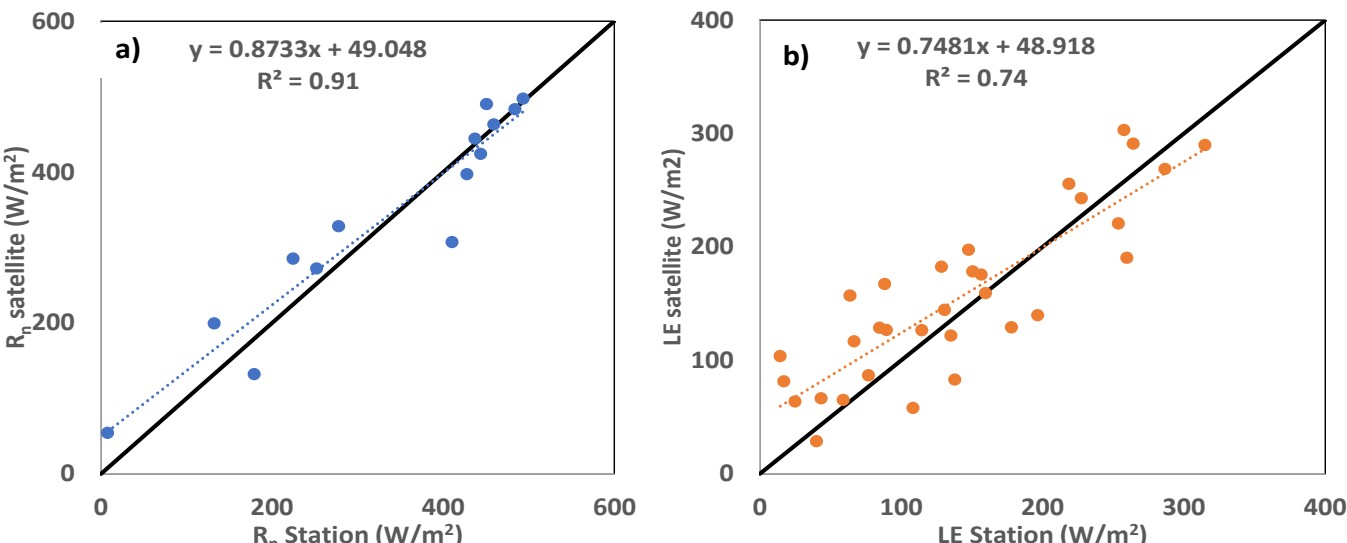

**Figure 3.** (**a**) Comparison of $R_n$ fluxes measured at the Borgo Cioffi field station for 14 different dates, with the $R_n$ fluxes retrieved from data acquired by Landsat-8 OLI/TIRS, after applying Equation (4). (**b**) The same comparison but for the LE fluxes, using data acquired at both sites and for all the 32 dates selected in this study.

**Table 1.** Statistical results for the ground–satellite comparison of Rn, S-SEBI LE, and daily ET.

|  | $R_n$ (W/m$^2$) | LE (W/m$^2$) | $ET_d$ (mm/d) |
|---|---|---|---|
| **$R^2$** | 0.91 | 0.74 | 0.72 |
| **StDev** | 5 | 8 | 0.2 |
| **RMSE** | 50 | 50 | 0.9 |
| **MAE** | 40 | 40 | 0.7 |
| **MBE** | −7 | −14 | −0.3 |
| **NSE** | 0.9 | 0.7 | 0.6 |
| **AI** | 0.997 | 0.996 | 0.996 |

It is worth noting that similar results were obtained when using the LST maps retrieved with the SW method proposed by [31], which follows the same algorithm used in this study (Equation (2)), but with different values for the $b_k$ coefficients. More studies with different SW LST algorithms or fiducial input data ($R_n$, G, or $\alpha_s$) would be required to reduce such bias and subsequently the associated uncertainties.

Finally, it has to be mentioned that our results, using the S-SEBI model and LST data retrieved with an SW algorithm applied on TIRS data at the two cropland sites, were in good agreement with those obtained in the previous eight related studies [6–13]. This fact showed also the good performance of the S-SEBI model, together with the reliability of the use of Landsat-8 SW-LST data, over European agricultural sites.

### 4.2. Comparison with ERA-5/Land Reanalysis Data

Figure 5 shows the comparison of the ERA5-Land TE reanalysis data with the S-SEBI $ET_d$ retrieved from Landsat-8 data for a total of 71 pixels from the 32 dates selected.

Statistical results related to the comparison shown in Figure 5 show a significant decrease of the correlation ($R^2 = 0.39$) between both datasets, with a StDev of 0.9 mm/d, but similar statistical uncertainties (RMSE= 1.1 mm/d, MAE = 0.9 mm/d, and MBE= −0.3 mm/d) compared to those obtained for the ground–satellite comparison of the previous section (see $ET_d$ in Table 1). Therefore, the similar statistical uncertainties confirm that S-SEBI $ET_d$ values retrieved from Landsat-8 SW LST data perform correctly both at regional and local scales.

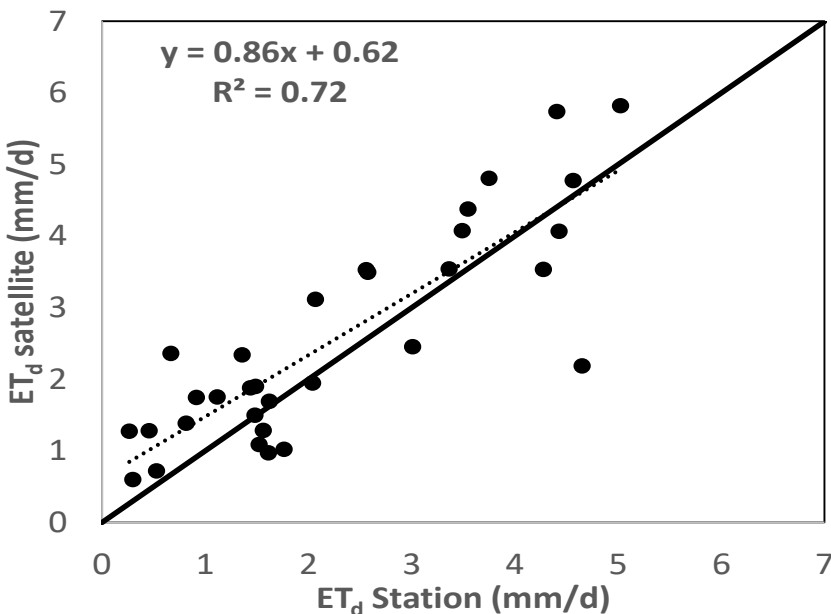

**Figure 4.** Comparison of daily evapotranspiration (ET$_d$) estimated from field data at both sites, with the ET$_d$ retrieved from data acquired by Landsat-8 OLI/TIRS, after applying the S-SEBI model.

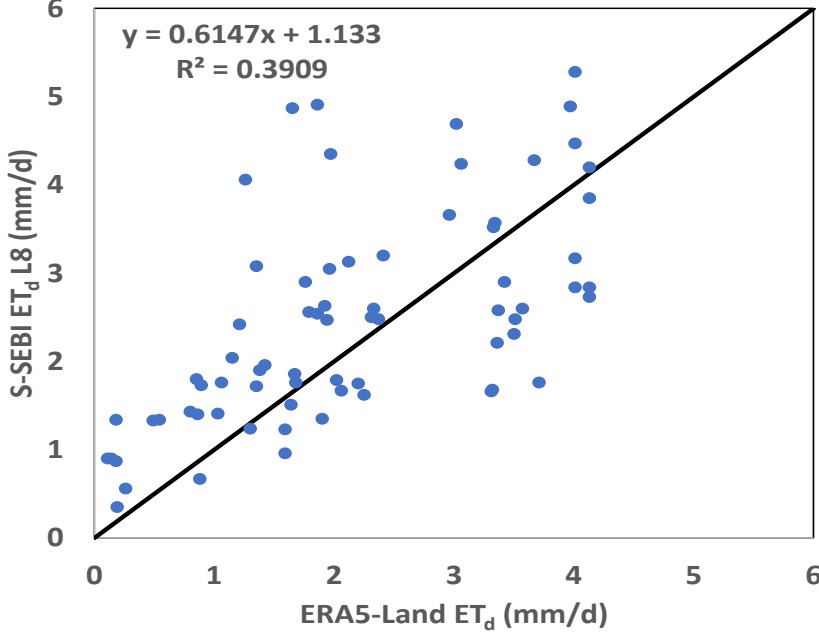

**Figure 5.** Comparison of total evaporation (TE) obtained from ERA5-Land reanalysis data and S-SEBI ET$_d$ values obtained from Landsat-8 inputs, after upscaling and geo-collocating the output scenes, all in mm/d.

## 5. Conclusions

In the present study, preliminary results of applying the S-SEBI ET model, fed with the official split-window LST and broadband albedos derived from Landsat-8 data at 100 m spatial resolution, showed it to be a suitable method when applied to cropland sites in the Mediterranean region, keeping in mind the good agreement achieved in relation to other studies around the world [6–13]. In terms of Rn and LE fluxes, as well as ET$_d$ values, the model results showed a good correlation with ground data ($R^2$ = 0.72–0.91), and uncertainties (StDev, RMSE, MAE) were within 5–50 W/m$^2$ and 0.2–0.9 mm/d, as expected by the ET community. A suitable G flux approximation is selected, accounting for the presence of vegetation. Additionally, high model indexes (NSE and AI) indicate

that S-SEBI could be an easy and fast method to estimate daily and high-spatial-resolution (100 m) ET values for worldwide extended crops, such as barley and corn. Compared at regional scale with the ERA5-land reanalysis data, our $ET_d$ results based on Landsat-8 data showed a poorer correlation (with respect to ground data) but a similar uncertainty (around 0.9 mm/d). These results encourage the achievements of future objectives for the ongoing research Tool4Extreme project, about the characterization of the hydrological cycle in the Mediterranean region as key to improve the water management in the context of global change.

**Author Contributions:** Conceptualization, V.G.-S. and R.N.; methodology, V.G.-S. and R.N.; software, V.G.-S.; validation, V.G.-S.; formal analysis, V.G.-S., R.N. and E.V.; investigation, V.G.-S.; resources, V.G.-S.; data curation, V.G.-S. and R.N.; writing—original draft preparation, V.G.-S., R.N. and E.V.; project administration, R.N. and E.V.; funding acquisition, R.N. and E.V. All authors have read and agreed to the published version of the manuscript.

**Funding:** This research was part of the R&D project PID2020-118797RB-I00 funded by MCIN/AEI/ 10.13039/501100011033 and PROMETEO-2021-016 project funded by the Generalitat Valenciana Government.

**Data Availability Statement:** Not applicable.

**Acknowledgments:** This work used eddy covariance data acquired and shared by the FLUXNET community, including these networks: AmeriFlux, AfriFlux, AsiaFlux, CarboAfrica, CarboEuropeIP, CarboItaly, CarboMont, ChinaFlux, Fluxnet-Canada, GreenGrass, ICOS, KoFlux, LBA, NECC, OzFlux-TERN, TCOS-Siberia, and USCCC. The FLUXNET eddy covariance data processing and harmonization was carried out by the ICOS Ecosystem Thematic Center, AmeriFlux Management Project and Fluxdata project of FLUXNET, with the support of CDIAC, and the OzFlux, ChinaFlux and AsiaFlux offices.

**Conflicts of Interest:** The authors declare no conflict of interest. The funders had no role in the design of the study; in the collection, analyses, or interpretation of data; in the writing of the manuscript, or in the decision to publish the results.

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
