# Peer review of "Evapotranspiration Retrieval Using S-SEBI Model with Landsat-8 Split-Window Land Surface Temperature Products over Two European Agricultural Crops"

_remotesensing, doi:10.3390/rs14112723_

Round 1
Reviewer 1 Report
I saw the authors implemented all my comments and I found this revised version of the work very easy to read and perfectly written. The concepts and the ideas communicated in this work are at high level. The manuscript has a good structure with good and coherent language. Therefore, I see that this version of the manuscript can be accepted for publication with RS-MDPI.
Author Response
Authors reply: Thank you very much for the previous reviewer’s comments, they clearly have improved the quality of the paper considerably.
Reviewer 2 Report
Thanks for the authors' effort to address the comments raised. However, there are still some points need to be further improved. I would like to recommend accepting this manuscript after minor revision.
1. Figure 3 looks a little blurry, I suggest the author replace it with a clear picture.
2. Figures 3 and 4 contain a lot of important information, but the author does not describe the content of the picture in detail, please improve.
3. Section 4.2, before using ERA5 for comparison, the author needs to describe the reason why ERA5 was used in the article. This makes the article read more logically.
Author Response
Thanks for the reviewer's suggestions, please find atached the answers to these comments.

Reviewer 3 Report
Generating daily ET maps from open-access imagery like Landsat-8 is a vital input for irrigation planning and other water budgeting-related studies. In this context, the output from this study has robust, practical applications. In my view, this work has shown good prediction performance with high practical applicability.
Minor comments:
1. Line 10 and 32 – “cropland” – Whether it is Crop and water consumption?
2. Line 15-16 requires clarity in the statement.
3. The results (lines 20-26) can come ahead of lines 16-20.
4. In Figure 2, the plot has some issues. A flat line after continuous fluctuations, especially in the Temperature readings. Please, recheck the plot.
5. All the figures require an increase in their font size and better representation.
6. ETd – maintain uniformity to keep the d in subscript in the write-up. In many places, subscript and superscripts are not appropriately followed.
7. Figure 5 missing equation and r2 value in the graph.
8. Modify Table 1 as per journal format.
I have listed only a few corrections, and many such typo errors need to be corrected throughout the manuscript. Overall, the manuscript requires a complete revision for the final version.
Author Response
Thanks to the reviewer for such valuables suggestions, please find attached a point by point answer to reviewer's suggestions.

This manuscript is a resubmission of an earlier submission. The following is a list of the peer review reports and author responses from that submission.
Round 1
Reviewer 1 Report
In this work, the authors studied Evapotranspiration (ET) retrieval using S-SEBI model with satellite data over two European agricultural crops. Compared with the site data, the ET data obtained by satellite inversion has a good consistency, which indicates the good performance of S-SEBI model and the reliability of satellite data. Moreover, due to the high temporal and spatial resolution of satellite data, ET retrieval using satellite data has great potential for the future application of hydrological cycle in Europe and even the world. The work is interesting and also can help us to better understand the land-atmosphere interactions under global climate change.
However, the results only evaluated the consistency between ET retrieve using satellite data and site data, without giving more basic features or details. Although this is a communication type article, it still needs to provide more information and evidence to support its own views and make the article complete. Here are some suggestions that need to be addressed before I can recommend accepting this manuscript:
- Lines 9-20, abstract must be improved. More research results should be presented instead of too much background introduction.
- It is recommended that the geographical locations of the two stations be drawn on a map. More basic climate characteristics (e.g., temperature, precipitation, and evaporation) about the two sites should be visualization and description in results.
- The authors describe that “Landsat-8 scenes covered the whole year at both site …” in line 116. However, only 34 Landsat-8 Collection 2 scenes were used in this study, please explain. If only 34 samples of satellite data can be matched to the two sites over a two-year period, that means there is no good time continuity in the same area.
- It is recommended to compare ET data retrieved by satellite with other Land reanalysis data (e.g., GLDAS, MERRA2 and ERA5/Land data), which have high temporal resolution in these regions. Besides, Taylor diagram are a good choice for evaluating different datasets.
- Please give full names for some abbreviations in the article (e.g., OLI and TIRS).
Reviewer 2 Report
The submitted research work aims to analyse the daily ET maps with the S-SEBI model. I found this paper very interesting but it does not hold the required amount of information to be published as a scientific paper in remote sensing journal. The undertaken research has not received any statistical validation of the obtained results. Moreover, the authors didn’t show sufficient explanation of the results and even the presented results are very modest. The level of research about analysing daily ET maps and the validation of LST Landsat products can not be compared with the level of this paper which is very low an contain only preliminary results. I see that this manuscript in its form and level DOES NOT deserve to be published with MDPI-RS and therefore the paper is REJECTED.